# Impact of an Antimicrobial Stewardship Strategy on Surgical Hospital Discharge: Improving Antibiotic Prescription in the Transition of Care

**DOI:** 10.3390/antibiotics12050834

**Published:** 2023-04-30

**Authors:** Alfredo Jover-Sáenz, Carlos Santos Rodríguez, Miguel Ángel Ramos Gil, Meritxell Palomera Fernández, Liliana Filippa Invencio da Costa, Joan Torres-Puig-gros, Dolors Castellana Perelló, Elisa Montiu González, Joan Antoni Schoenenberger-Arnaiz, Juan Ramón Bordalba Gómez, Xavier Galindo Ortego, María Ramirez-Hidalgo

**Affiliations:** 1Unidad Territorial Infección Nosocomial (UTIN), Hospital Universitari Arnau de Vilanova de Lleida (HUAV), Institut de Recerca Biomèdica (IRBLleida), 25198 Lleida, Spain; 2Servicio de Farmacia, Hospital Universitari Arnau de Vilanova de Lleida (HUAV), 25198 Lleida, Spain; 3Servicio de Urología, Hospital Universitari Arnau de Vilanova de Lleida (HUAV), 25198 Lleida, Spain; 4Servicio de Otorrinolaringología, Hospital Universitari Arnau de Vilanova de Lleida (HUAV), 25198 Lleida, Spain; 5Departament de Salut Pública, Universitat de Lleida (UdL), 25006 Lleida, Spain

**Keywords:** antimicrobial stewardship, use antimicrobials, care transition, discharge prescribing, security of care

## Abstract

Antimicrobial stewardship programs (AMSPs) are essential elements in reducing the unnecessary overprescription of antibiotics. Most of the actions of these programs have focused on actions during acute hospitalization. However, most prescriptions occur after hospital discharge, which represents a necessary and real opportunity for improvement in these programs. We present an AMSP multifaceted strategy implemented in a surgical department which was carried out by a multidisciplinary team to verify its reliability and effectiveness. Over a 1-year post-implementation period, compared to the pre-intervention period, a significant reduction of around 60% in antibiotic exposure occurred, with lower economic cost and greater safety.

## 1. Introduction

The increase in bacterial resistance to antimicrobials has led to the creation of antimicrobial stewardship programs (AMSP) [1,2,3,4]. Historically, AMSP has proved to be highly effective in hospital facilities and has been implemented in a community setting recently [5,6,7].

Although AMSP interventions to improve antimicrobial use in hospitalized patients are well-established [8], the interface between hospital and community remains challenging and exposes patients to potential harm. In general, hospital AMSP teams interfere in antibiotic prescribing through education, audits, and informative feedback to the prescriber. The program includes various educational and training actions, highlighting the adjustment of treatment suggested by consensus guidelines, the reduction in consumption, and the early enteral conversion during admission, thus reducing the risk of complications and promoting the reduction in healthcare-associated infections [9,10]. However, such interventions have not been designed to improve antimicrobial prescribing at hospital discharge. Recent studies suggest that between 30% and 40% of antibiotics prescriptions associated with a hospital stay occur after discharge [11,12] in more than 1 in every 8 patients [13], with inadequate suitability 50–70% of the time, either for a longer duration than necessary or for abuse of fluoroquinolones (FQ) [14,15,16,17]. This proportion could be higher in surgical services, specifically urology [18]. A global surveillance study on urological infections that recruited patients from more than 70 countries showed that more than half of hospitalized urological patients received antibiotics, and 46% of them were broad-spectrum [19].

As AMSPs continue to evolve, novel approaches are needed to provide extended integrated care focused on antimicrobial therapy at discharge. However, the impact of interventions targeting this fact is unknown, and there is little evidence published so far [20]. In 2019, the Centers for Disease Control and Prevention (CDC) recognized the importance of optimizing antibiotic therapy at discharge in a hospital AMSP [21]. However, methods to improve the use of antibiotics at discharge are not well developed.

The multi- and interdisciplinary nature of AMSP teams and their interaction may be a good strategy to better define antibiotic therapy in transitions from inpatients to other settings. Experiences in the reconciliation of chronic medications, carried out by pharmacists at the time of discharge, have proven to be effective in promoting patient education and reducing errors and readmission rates, which would help facilitate the improvement in antimicrobial use [22,23]. This aspect, combined with the performance of the infection control team (infectologists and nurses), can favor the process. Moreover, the role of this group can be vital in the delivery of the exact number of tablets for the completion of treatment. The standard marketing of community antibiotic packages usually exceeds the number of tablets needed for a usual treatment, favoring the accumulation of medication in their home and future misuse; therefore, control in this action would also serve as a measure of safety and economic savings. A recent report by the Cooperative for the Distribution of Medicines, Health Products, and Services COFARES reports that 60.1% of Spaniards keep antibiotics left over from previous treatments for emergency cases [24].

Our work aims to measure the impact of a multimodal antimicrobial reconciliation intervention at discharge on the consumption, suitability, duration, safety, and cost of treatment in a urology service.

## 2. Materials and Methods

### 2.1. Design, Setting, and Study Periods

A semi-experimental comparison study was conducted before and after an intervention. The study period included 12 months, between March 2021 and February 2022, compared to a period between 2018–19, where the AMSP actions of reconciliation at discharge had not been developed. The study was carried out in the urology department (22 beds) of a 400-bed General University Hospital belonging to the public health network of Catalonia (CatSalut), Spain, with a reference population of 450,000 inhabitants. This service has an average monthly admission of 150 patients (about 30% are infections). Since 2012, the hospital has had an interdisciplinary AMSP team consisting of doctors and nurses trained in infectious diseases, mostly belonging to the Department of Nosocomial Infection Control (DNIC), together with professionals from various specialties (hospital pharmacy, microbiology, internal medicine, intensive, and preventive medicine). The hospital AMSP has institutional recognition and is included in a regional AMSP project, constituted by professionals from different specialties and groups, which encompassed other areas (primary care, long-term care facilities, and geriatric residences) called P-ILEHRDA. The design was created considering the consensus AMSP document published by the Spanish Society of Infectious Diseases and Clinical Microbiology [9], adapted to the characteristics of the region and the centers. This model has been described in previous publications [25,26]. In 2021, this team initiated a pilot plan in the urology department after mutual agreement, participation, and acceptance, based on the adequacy in the management of vascular catheters, commitment to the early removal and promotion of enteral conversion of medication in the first 48–72 h from patient admission. At the same time, the association of AMSP actions aimed at the adequacy of prescriptions made during hospitalization and an antimicrobial reconciliation at discharge. These actions were carried out by members of the DNIC and hospital pharmacists.

### 2.2. AMSP Actions

The program included the following actions:-During hospitalization: 1. Developing and updating antibiotic treatment protocols of the most prevalent urinary infections, based on scientific evidence and local sensitivities, available on the mobile application (ProAPP Lleida) and intranet of the institution; 2. Sharing of AMSP actions with the urology department; 3. General and specific training of professionals; 4. Daily review of all positive microbiological results (blood cultures and any other samples), except weekends and holidays; 5. Daily written non-imposed advice for professionals in computerized SAP “Systems, Applications, Products in Data Processing” medical history, advice on-site or by telephone. The actions could take place in relation to any positive microbiological result and/or systemic antibiotic prescription made for admitted patients. 6. Promotion of enteral conversion at 48–72 h on possibilities and characteristics of the microorganism and oral bioavailability of the antibiotic, with clinical stability and respected digestive tract.-At the time of discharge: 1. Advice on the suitability of empirical or targeted oral antimicrobial therapy, dose, frequency, and duration; 2. Assess the absence of interactions, duplications, and allergy/intolerance; 3. Delivery of a recommended number of antibiotic tablets, submitting oral and written information to the patient; 4. Computerized notification of the reconciliation to the community primary care physician; 5. Periodic feedback of results to the team members. The flowchart is shown in Figure 1.

No restrictive measures were made to prescriptions. Adherence to the recommendations was evaluated at 24–48 h from intervention. The information was collected prospectively 30 days after discharge, and the degree of compliance, readmission, reconsultation, and/or retreatment, and mortality was also counted. Likewise, the appearance of diarrhea due to *Clostridioides difficile* was determined on day 90 from hospital discharge.

### 2.3. Measurement of Consumption and Economic Impact

The primary outcome of the study was to analyze the change in global antimicrobial consumption at discharge, stratified by complicated non-surgical infectious entities (prostatitis/epididymo-orchitis, catheter-associated urinary tract infection), surgical complicated (post-surgical or associated with an invasive technique -biopsies, catheter implantation-) and uncomplicated (pyelonephritis and miscellaneous) in relation to the prescription of non-recommended antibiotic groups (NRA) (amoxicillin-clavulanate, 3rd generation cephalosporins and FQ) versus recommended antibiotics (RA) (co-trimoxazole, cefuroxime axetilo, fosfomycin trometamol, amoxicillin), before and after AMSP actions that were carried out in year 2018 (first period without intervention) and 2021 (second period with intervention). The secondary outcome was to analyze the mortality, readmission, retreatment, and drug adverse effects (AE) based on the incidence of *C. difficile* infection 30 days from the intervention. The third outcome was the reduction in expenses attributable to the results of the AMSP reconciliation.

### 2.4. Evaluation Methods and Sources of Information

To evaluate the consumption of antimicrobials, the methodology of the CDC National Healthcare Safety Network was used and expressed as the number of Days of Antimicrobial Therapy (DOT). DOT is defined as the aggregate sum of days for which any amount of a specific antimicrobial agent was administered to individual patients, regardless of the administered dose. This parameter has greater clinical relevance, requires individual patient data, is not modified by dose adjustment, and can be applied to both the pediatric and adult population. In our case, it was calculated with the sum of days in which any amount of an antimicrobial agent was administered to a patient dispensed at discharge, according to the defined days.

The Pharmacy Service evaluated the consumption data obtained from the specific electronic prescription software used in the health system called Silicon^®^, integrated into SAP-ARGOS (Electronic Health Records (EHR), together with the cost of antimicrobials (expressed in euros -€-)), according to the standard fee in the study periods. The average stay, discharge, mortality, and readmissions-related data were provided by the hospital’s technical registry. The clinical resolution was assessed only in patients with available follow-up data, defined as the resolution of signs and symptoms of the infection at 30 days with no requirement for further antimicrobial therapy other than scheduled. Informed consent of each patient was obtained. The appearance of *C. difficile* infection, as an index of quality of care, was identified by the microbiological surveillance system of the health area.

### 2.5. Statistical Analysis

Continuous quantitative variables were expressed as mean +/− standard deviation (SD), and categorical variables as frequencies and percentages (%). The Odds Ratio (OR) was used as an association measure, and the difference in percentages or means between the pre- and post-intervention periods as an impact measure. These data were presented with 95% confidence intervals (CI). Chi-square and Fisher’s exact test were used to contrast two qualitative variables, and the Student–Fisher *t*-test was used for the comparison of means between both periods. Significance was defined as *p* < 0.05. Data were analyzed using the statistical programs IBM SPSS Statistics (version 22) and Epidat (version 3.1) of the Pan American Health Organization.

In the assessment of the appropriate sample size, the applied statistical tests managed to detect differences of less than 15% between both groups with an alpha risk of 5% and a power of 80%.

### 2.6. Ethical Declaration

This project was approved by the Research Ethics Committee of the Arnau de Vilanova University Hospital (CEIC-2545). Informed consents were subject to compliance with Organic Law 3/2018, of December 5, Protection of personal data and Guarantee of digital rights and Regulation (EU) 2016/679 of the European Parliament and the Council of 27 April 2016 on Data Protection.

## 3. Results

Of the 1450 patients evaluated, 466 (32.1%) presented a urological infection. Of these, 259 were included in the final phase of the study: 143 in the pre-intervention period and 116 in the post-intervention period (Figure 2). The demographic and clinical characteristics of the patients are shown in Table 1. The mean (SD) Charlson comorbidity index score was 1.5 (1.8) (*p* = 0.030); There were no significant differences in comorbidity conditions. Complicated non-surgical urinary tract infection prevailed over other infections (139 (53.9%)), with prostatitis being the most common diagnosis (87 (62.6%)).

During the intervention period, 573 interventional advices were made at discharge. Overall compliance with the antimicrobial treatment protocol was 83.0% (385/464). The degree of acceptance of advice at discharge was 89.4%, being the duration of the antibiotic according to the infectious entity, the most accepted intervention 79.5% (OR 2.5, (95% CI, 1.2 to 6.3), *p* = 0.035), followed by the frequency of administration 61.4% (OR 3.9, (95% CI, 0.9 to 20.7), *p* = 0.048). Specifically, according to the type of antimicrobial, the prescription of cefuroxime axetil was the most accepted (OR 5.7, (95% CI, 2.6 to 12.5), *p* < 0.001).

Table 2 shows the typology of antimicrobials prescribed according to study periods. The primary endpoint, optimal antimicrobial prescribing at discharge, was associated with the implementation of the intervention in 62 patients out of 116 (53.4%) versus 60 out of 143 (42.0%); *p* = 0.043. The absolute increase in optimal prescribing in the post-intervention group was consistent with the prescription of any of the RA (*p* = 0.005) and FQ avoidance (OR 0.44, (95% CI, 0.22 to 0.87), *p* = 0.016). The prescription adjustments according to RA by entities are reflected in Figure 3.

Antimicrobial prescription remained associated with the primary outcome, and patients in the post-intervention period were 0.27 times (21.5%) (95% CI, −1.50 to 39.28) likely to be prescribed an optimal antimicrobial regimen at discharge. The factors that contributed the most to the improvement of optimized prescribing at discharge were: reductions in prolonged duration expressed in DOT (18.7 vs. 11.5 (38.5%); mean difference, −7.2 days (95% CI, −8.6 to −5.7); *p* ≤ 0.001) and treatment with a smaller number of tablets according to commercial packaging (4.4 vs. 6.0 (26.3%); mean difference, −1.4 tablets (95% CI, −2.4 to −0.3), *p* = 0.009). No statistically significant differences were observed in clinical resolution, readmission, retreatment, or mortality at 30 days (Table 3). The intervention was associated with fewer AE of medication, presented in only six cases in the pre-intervention period (4.2%) versus 0% in the post-intervention, mainly due to reductions in *C. difficile* infection by day 90, although not significant.

A total of 708,558 prescriptions were made in the 259 patients corresponding to 4000 DOTs, of which 2761 (69.0%) occurred after hospital discharge. NRAs were the most common antimicrobial prescription (52.9%). The combined mean DOT for both periods during hospitalization was 4.8 (SD; 2.8), while the mean DOT outside hospitalization was 10.7 (SD; 6.6). The intervention was associated with a decrease in the total duration of the antimicrobial therapy. In the post-intervention period, DOT at discharge was significantly reduced by 42.9% (absolute difference −7.2, (95% CI, −6.11 to −8.79), *p* < 0.001 days of antibiotic), suffering an increased reduction of 1/3 greater when there the exact tablets delivery was detailed (16.7 vs. 6.53). Table 4 shows the DOTs in both study periods, according to hospital admission or after discharge, and cost savings. DOTs for complicated non-surgical urinary tract infection were reduced (adjusted absolute difference −7.1, (95% CI, −5.0 to −9.2), *p* < 0.001 days of antibiotic) in the uncomplicated urinary tract (adjusted absolute difference −5.4, (95% CI, −3.4 to −7.4), *p* < 0.001 days of antibiotic) and in surgical complicated urinary tract infection in the post-intervention period (adjusted absolute difference −5.0, (95% CI, −0.6 to −9.5), *p* = 0.029 days antibiotic). The most significant reductions in DOT were obtained in prostatitis (adjusted absolute difference −6.9, (95% CI, −4.0 to −9.7), *p* < 0.001 days of antibiotic), pyelonephritis (adjusted absolute difference −5.5, (95% CI, −3.2 to −7.7), *p* = 0.001 days of antibiotic) and device-associated infection (adjusted absolute difference −5.7, (95% CI, −3.3 to −8.0), *p* < 0.001 days of antibiotic). There were no reductions of DOT in the remaining entities.

The average annual cost of antibacterials between periods, according to minimum commercial packaging, decreased from €7.4 to €6.0 (mean difference −1.4, (95% CI, −2.4 to −0.3), *p* < 0.001) and in the post-intervention period between minimum packaging and equivalence with the grouped tablets saved, according to duration of treatment, it was reduced by 27.7% from €6.0 to €4.4 (average difference −1.5, (95% CI, −4.9 to 3.8).

## 4. Discussion

The results of this study suggest that the multifaceted implementation of an antimicrobial reconciliation at the time of hospital discharge, carried out jointly by diverse professionals belonging to a clinical advisory team, was associated with a more suitable prescription in a surgical service. Moreover, the involvement of pharmacists in this equation favored safety by promoting health education and patient monitoring, avoiding both the AE of an inappropriate prescription, as pointed out by various studies [27,28], and the inappropriate accumulation of surplus drugs.

The use of antimicrobials in surgical specialties is a potential target for AMSPs, as large quantities of antimicrobial agents are prescribed in those departments. Urologists make an average of 6 million prescriptions per year for this concept in the United States [29], ranking eighth among all specialties in antimicrobial prescription in 2015 [30]. This situation benefits not only the patient but also the professionals. Multifaceted interventions targeting urologists that include recommendations, training, and feedback, could have a positive impact by optimizing the immediate use of antibiotics and promoting changes in their prescribing practice when they feel involved [31,32]. Recent evaluations by Srinivasan et al. [33] of the surgical prescriber’s perception of knowledge and use of antibiotics suggest not only that there is an opportunity for improvement but also a general desire for further education.

CDC highlighted in 2019 the importance of antimicrobial administration in the transition from hospital discharge as an opportunity to improve prescribing, especially in UTIs [21]. Vaughn et al. [34] identified an over-prescription of close to 50% in a cohort of 25,000 patients with UTIs. In line with our results, the observed findings were largely due to both the drug choice and the excessive duration of therapy, two of the “4 moments” identified as key to improving antibiotic prescription in the patient’s transition [35].

In the first aspect, the use of FQ is common in both inpatients and outpatients, and there are numerous examples of successful strategies for the reduction in this group of antimicrobials in any of these areas. An important contribution of our study is to highlight this possibility of improvement at the time of the transition that is often dismissed [16]. In the development of a pharmacological regimen, there are several opportunities for selecting an optimal antibiotic. While several antimicrobials may be correct in one indication, efforts to choose the best one can improve their use and minimize adverse consequences such as the development of resistance, a risk that may persist beyond immediate treatment. A recent case-control study [36] evaluating FQ resistance in *Escherichia coli* found that receiving at least one FQ prescription before resistance was known was associated with an increased risk of FQ-resistant *E. coli* colonization or infection. This risk was highest in the first year after taking the antibiotic (OR 2.67), decreasing progressively until it was undetectable two years later (OR 1.09). Decreasing FQ use has become a priority in health approaches due to the ecological impact of these antibiotics and recent safety concerns [37,38]. In our health area, an AMSP has been developed in the community that has led to a significant reduction in the prescription rates and FQ resistance of the most common Enterobacteriaceae over the past 5 years [39].

Although any antibiotic is susceptible to inducing *C. difficile* infection [40], it has been suggested in multiple studies that cephalosporin use is independently associated with this fact [41]. However, this aspect seems specific to 3rd generation cephalosporins. Some studies have mentioned how 1st and 2nd generations cephalosporin regimens in surgical processes do not seem to show increased incidence or risk of *C. difficile* infections [42,43], which would allow its use as a reasonable alternative in enteral conversion, mainly when alternative options are limited by resistance or AE [44], as long as they are not used in the management of prostate infections where they have a poor diffusion [45].

In our work, a prolonged frequency of antimicrobials at discharge (38.5%) before the intervention was observed, which makes it one of the main actions in antimicrobial reconciliation [13] as shorter durations become more widely accepted [46]. Unnecessarily prolonged courses increase the patient’s risk of AE and *C. difficile* infection, in addition to the overall risk of resistance development.

The duration of therapy has been repeatedly identified as the situation where there are greater differences between the recommendation of the guidelines and the reality of prescribing, especially in the transition of health care. In a study [47] aimed to determine the adequacy of prescriptions issued at discharge from a 300-bed academic hospital, from 236 prescriptions, only 21% were appropriate in duration, and the vast majority (71%) exceeded the recommended guidelines. These findings mirror the observed in other evaluations. Brower et al. [48] reported that 81% of patients discharged with antibiotics for UTIs or pneumonia received an excessive duration of the prescription, with an overall mean of 4 days longer than recommended in guidelines.

Although it was not the objective of this study, there are certain possibilities that justify this procedure. A systematic review of 35 studies that sought to understand the factors of this behavior found that complacency in meeting the patient’s perceived expectations of the antimicrobial and the physician’s fear of patient complications were the main ones [49]. Likewise, there may be a perception of the patient’s vulnerability at the time of discharge due to their lack of monitoring at follow-up, causing clinical decisions to be mediated. In this situation, clinicians may err in prescribing longer antibiotic regimens as a precaution to avoid readmission without weighing the concomitant short- and long-term risks of inappropriate antibiotic use [50].

The measurement of the consumption and typology of antimicrobials is a common quality action in hospitalized patients, but it is rarely carried out after discharge. Their determination is an opportunity to establish strategies aimed at improving prescriptive practices through reconciliation [51]. On the other hand, it can be an associated advantage in community safety with the detailed delivery of the exact tablets for completion of antimicrobial treatment. This aspect minimizes the risk of having remaining antibiotics at home, which favors future self-treatment, as reflected in European studies [52,53]. Spain, unlike other members of the European Community, is subject to the fixed marketing format, often not adjusted to the current trend and recommendations [54]. In countries where antibiotics are dispensed in this way, there is a four- to five-fold increased risk of stockpiling residual antibiotics and being given to others, increasing the possibility of inducing resistance [55].

In our study, we have been able to verify the existence of the final antimicrobial consumption of an added reduction of 1/3 to that already provided by the actions of the reconciliation. This fact has not been described or quantified economically in the literature and provides greater value to this study.

Our study has some limitations. Firstly, the high frequency of optimal prescription of antimicrobials at discharge, around 70%, in the post-intervention group. The previous existence of a territorial transversal institutional AMSP could strengthen the knowledge and prescriptive behavior of urologists in the post-intervention period. Secondly, it has been described that each day of excess antimicrobial has been associated with a 9% higher chance of developing an AE [56]. In our interventions, although we had fewer complications, there was not a significant reduction in severe AE, readmissions, or mortality, otherwise described by Vaughn et al. [9], perhaps because of the small sample size. Finally, the reduced presence of prostate infections in the intervention period, inversely to pyelonephritis, was attributed to the peak period of the COVID pandemic, which could select the type of admission by restricting the transfer from nursing homes to the hospital of elderly patients susceptible to prostate pathology and involuntarily influencing the duration of some treatments.

## 5. Conclusions

The antimicrobial reconciliation in the transition from hospital discharge as a main AMSP action is an opportunity that should not be missed. The results of this study show how the strategy carried out by a multidisciplinary team is an effective measure in antibiotic optimization that can reduce by around 60% the possibility of antibiotic exposure, with a lower economic cost and greater safety.

## Figures and Tables

**Figure 1 antibiotics-12-00834-f001:**
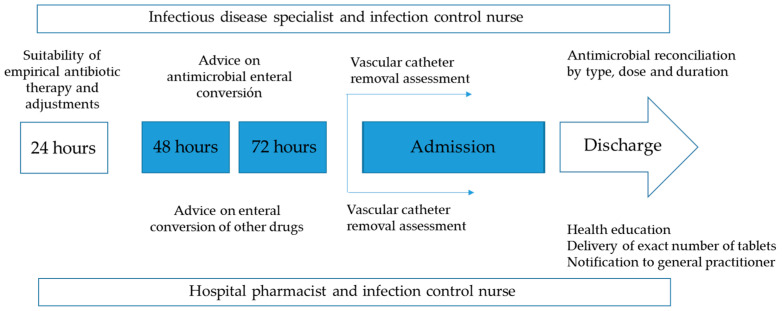
Implementation strategy.

**Figure 2 antibiotics-12-00834-f002:**
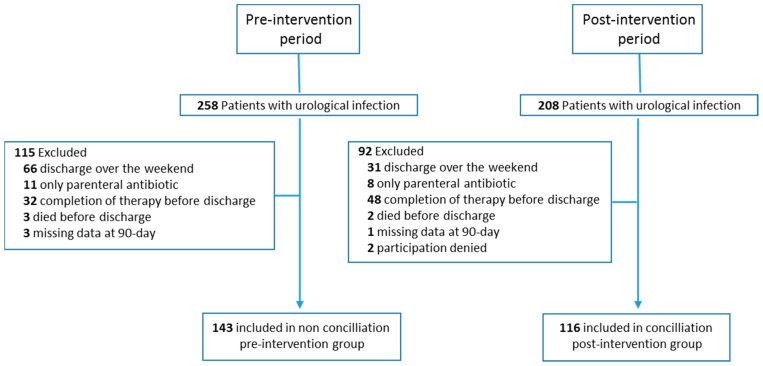
Flowchart of study.

**Figure 3 antibiotics-12-00834-f003:**
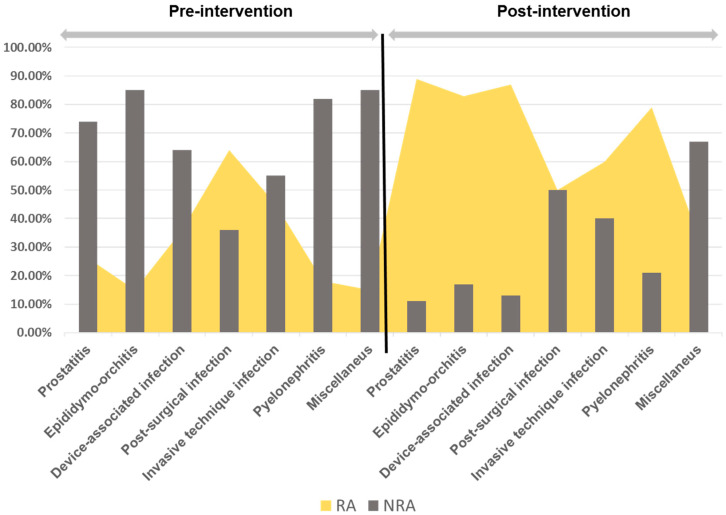
% Compliance with recommended antimicrobial (RA) (cotrimoxazole, cefuroxime, amoxicillin, fosfomycin-trometamol) or non-recommended (NRA) (FQ, third-generation cephalosporins, co-amoxiclav) according to infectious entity.

**Table 1 antibiotics-12-00834-t001:** Patient demographics and clinical entity features.

	All Cases (*n* = 259) N° (%)	Group Pre-Intervention (*n* = 143) N° (%)	Group Post-Intervention (*n* = 116) N° (%)	*p* Value
Mean age (SD), year	65.5 (16.8)	64.4 (15.1)	61.6 (16.1)	NS
Male gender	213 (82.2)	132 (92.3)	81 (69.8)	<0.001
Comorbidity index				
Charlson ≥ 2	103 (39.8)	62 (46.4)	41 (35.3)	NS
Length of hospital stays, mean (SD), days	4.9 (3.1)	4.7 (2.7)	5.1 (3.5)	NS
Infectious urinary entity and group				
Prostatitis	87 (33.6)	69 (48.2)	18 (15.5)	<0.001
Epididymo-orchitis	23 (8.9)	17 (11.9)	6 (5.2)	NS
Device-associated infection	29 (11.2)	14 (9.8)	15 (12.9)	NS
Post-surgical infection	19 (7.3)	11 (7.7)	8 (6.9)	NS
Invasive technique infection	16 (6.2)	11 (7.7)	5 (4.3)	NS
Pyelonephritis	74 (28.6)	17 (11.9)	57 (49.1)	<0.001
Miscellaneous	10 (3.9)	4 (2.8)	6 (5.2)	NS

Unless otherwise indicated, data are expressed as number (%) of patients. NS: not significant.

**Table 2 antibiotics-12-00834-t002:** Antimicrobial prescription.

	Group Preintervention *n* (%)	Group Postintervention *n* (%)	Absolute Difference 95% CI	*p* Value
Quinolones (J01M)	37 (25.9)	14 (12.1)	−13.8 (−25.9 to 3.8)	0.016
Third-generation cephalosporins (J01DD)	31 (21.7)	23 (19.8)	−1.0 (−11.8 to 8.1)	NS
Co-amoxiclav (J01CR02)	15 (10.5)	17 (14.7)	4.2 (−4.0 to 12.3)	NS
Cotrimoxazole (J01EE01)	44 (30.8)	26 (22.4)	−8.4 (−19.1 to 2.4)	NS
Cefuroxime (J01DC02)	9 (6.3)	32 (27.6)	21.3 (12.2 to 30.5)	<0.001
Amoxicillin (J01CR02)	2 (1.4)	1 (0.9)	−0.5 (2.0 to −3.1)	NS
Fosfomycin trometamol (J01XX01)	5 (3.5)	3 (2.6)	−0.9 (−5.1 to 3.3)	NS
**Total recommended antibiotics (RA)**	60 (42.0)	62 (53.4)	11.5 (−0.7 to 23.6)	0.043

Unless otherwise indicated, data are expressed as number (%) of patients. Co-amoxclav: amoxicillin–clavulanic acid; RA: recommended antibiotics (cotrimoxazole, cefuroxime, amoxicillin, fosfomycin-trometamol); NS: not significant.

**Table 3 antibiotics-12-00834-t003:** Patient outcomes.

Patients n° (%)	Group Pre-Intervention *n* (%)	Group Post-Intervention *n* (%)	Absolute Difference 95% CI	*p* Value
30-day overall mortality	2 (1.4)	1 (0.9)	−0.5 (−3.1 to 2.2)	NS
30-day readmission	19 (13.3)	16 (13.8)	0.5 (−7.9 to 8.9)	NS
30-day Retreatment/emergency or community visit	22 (15.4)	8 (9.5)	−5.9 (−13.9 to 2.1)	NS
No clinical resolution	17 (11.9)	10 (8.7)	−3.3 (−10.6 to 4.1)	NS
Adverse drug event	3 (2.1)	0 (0)	−2.1 (−5.1 to 1.2)	NS
90-day *C. difficile* infection	3 (2.1)	0 (0)	−2.1 (−5.1 to 1.2)	NS

Unless otherwise indicated, data are expressed as number (%) of patients, NS: not significant.

**Table 4 antibiotics-12-00834-t004:** Days of antimicrobial therapy (DOTs), DOTs avoided, and cots.

	Group Pre-Intervention (*n* = 143)	Group Post-Intervention (*n* = 116)	Absolute Difference 95% CI	*p* Value
DOT in	4.63	4.95	0.88 (−0.38 to 1.00)	NS
DOT out	14.01	6.56	−7.45 (−8.73 to −6.17)	<0.001
DOT in + out	18.67	11.54	−6.16 (−8.66 to −5.60)	<0.001
Mean DOT out by UD	16.74	9.56	−7.21 (−8.61 to −5.81)	<0.001
Mean tablets out by UD	26.00	13.46	−12.54 (−15.64 to −9.43)	<0.001
Mean tablets saved	5.13	5.29	0.16 (−1.08 to 1.40)	NS
Mean DOT saved	2.73	3.00	0.27 (−0.44 to 0.97)	NS
Average cost per MPM (€)	7.42	6.02	−1.40 (−2.44 to −0.35)	0.009
Average cost saved (€)	1.11	1.66	0.56 (0.92 to 0.19)	0.003

UD: Units dispensed according to the MPM; MPM: minimum package marketed; €: euros; NS: not significant.

## Data Availability

Not applicable.

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
