# Peer review of "Impact of an Antimicrobial Stewardship Strategy on Surgical Hospital Discharge: Improving Antibiotic Prescription in the Transition of Care"

_antibiotics, 2023, doi:10.3390/antibiotics12050834_

Round 1

Reviewer 1 Report

The present manuscript by Alfredo Jover-Sáenz is scientifically outstanding and excellently written. It shows important aspects of antimicrobial stewardship. From my point of view, no corrections are necessary. As it is one of the best manuscripts I have read this year, I recommend it for publication as it is. The readers of Antibiotics Basel and the scientific community will benefit greatly from this. Congratulations! 

Author Response

Dear reviewer,

We sincerely appreciate all of the valuable comments of our work, thank you very much!

Reviewer 2 Report

The study shows interesting findings and improve the knowledge in the field. The paper deals with the topic of actuality. The paper is clear, well written, and the organization is very good.The references are up to date, and they are well organized according to the format required by the journal.

Author Response

Dear reviewer,

We appreciate you for your precious time in reviewing our paper. Thank you very much for your valuable comments!

Reviewer 3 Report

for the manuscript entitled "Impact of an antimicrobial stewardship strategy on surgical hospital discharge. Improving antibiotic prescription in the transition of care", I can suggest that It can be acceppted for the publication after checking its english.

Author Response

Dear reviewer,

We sincerely appreciate all of the valuable comments of our work. We have revised our manuscript and the language have been checked according to your suggestion.